# Research on the key technology of high-precision waveform synchronization identification and measurement

**Dan Gui**[1], **Zong-jie Yu**[2]*

**1** School of Electronic Engineering, Wuhan Vocational College of Software and Engineering, Wuhan, China,
**2** Research & Development Department, FiberHome Telecommunication Technologies Co., Ltd, Wuhan, China

* zjyu@fiberhome.com.cn

**Data Availability Statement:** All relevant data are within the manuscript and the supporting information- compressed/ZIP File archive.

**Funding:** the industry-University-Research Project of Wuhan Municipal Education Bureau

## Abstract

High-precision waveform identification and measurement are effective for waveform detection and evaluation in signal processing. The accuracy of waveform identification, precision of measurement, and speed of response are important indicators of waveform measurement instruments. To detect the waveform accurately, a hold and attenuation circuit divided into two is designed, and the STM32F4 microcontroller is used to accurately capture and perform spectrum analysis using a high-precision analog-to-digital converter based on fast Fourier transform technology to identify key parameters, such as waveform type, frequency, peak-to-peak value, and duty cycle. To improve the recognition accuracy and response speed, technical solutions, such as high-frequency sampling and over-zero detection, are used to improve the system efficiency. Algorithm simulation, circuit simulation, and physical testing show that the high-precision waveform synchronization recognition circuit and algorithm can accurately recognize various essential waveforms in the voltage and frequency ranges of $50\,mV \leq V_{PP} \leq 10\,V$ and $1\,Hz \leq f \leq 50\,kHz$, respectively, and simultaneously measure important parameters, such as frequency, peak-to-peak value, and duty cycle with an accuracy within ±1%. Intelligent linkage, no intermediate parameter setting, and a response speed of approximately 0.3 s make it suitable for such applications as fast and high-precision waveform intelligent detection and display. The method is highly integrated, simple to operate, cost-effective, and practical.

## Introduction

Waveform recognition technology is important for the wide application of digital instrumentation [1, 2]. With the increasing demand for accuracy and response speed of waveform recognition, high-precision waveform recognition, fast display, and accurate measurement of key parameters are essential in intelligent troubleshooting, clinical electrocardiography detection, and radar waveform recognition [3–5]. At present, waveform recognition and measurement typically use waveform coefficient recognition, calculation of the root mean square (RMS)

(CXY202014); Education Research Program Project of Hubei Provincial Department (B2020405).

**Competing interests:** The authors have declared that no competing interests exist.

value, and other methods that can adapt to the most common measurement scenarios [6, 7]. However, cases with low frequency and high voltage amplitude suffer from inaccurate waveform recognition and large peak-to-peak ($V_{pp}$) errors [8]. Moreover, owing to the low-frequency measurement method used in the low-frequency case, the sampling rate is slow, resulting in a long response time, which seriously affects further applications of the relevant technology [9].

Periodic waveform measurements are usually divided into two categories: sinusoidal and non-sinusoidal signals. Sinusoidal signals are comparatively easy to identify mainly because the ideal sinusoidal RMS and fundamental RMS are equal, the high harmonics are zero, and the proportionality between the RMS and the peak-to-peak value is fixed [10]. The identification method for non-sinusoidal waves is relatively complex. One method is to calculate the RMS value of the periodic signal and use the relationship between the RMS value and the peak value for identification; however, the accuracy of this method is relatively insufficient. Another method is to use the Fourier series to analyze the base wave characteristics of the signal using the relationship between the base wave RMS and the total RMS, which is more accurate; however, the realization of the difficulty is more complex [11, 12]. As an important method for waveform parameter measurement, the lock-in frequency measurement method can determine frequencies with a slow drift over a small frequency range with high precision and efficiency [13].

Traditional waveform recognition and measurement instruments commonly have insufficient amplitude span and small frequency gaps. In ordinary experiments and measurement environments, the approach can still be used normally, but for amplitude and frequency spanning with large scales (50 mV ≤ $V_{pp}$ ≤ 10 V, 1 Hz ≤ f ≤ 50 kHz), the shortcomings are obvious. Waveform analysis at low input voltages and high input frequencies is prone to such issues as inaccurate signal analysis, insufficient precision (>1%), and slow response speed (>3 s). Furthermore, the cumbersome instrument connection and analysis process requires a certain degree of professionalism from the user, and its practicality is greatly reduced for staff with a relative lack of professional foundation [14]. Therefore, the development of a cross-scale, high-precision, fast-response waveform recognition and measurement device that can achieve a variety of waveform recognition and real-time measurement tasks of important parameters is significant for observing and analyzing signals in practical applications.

Considering the practicality, development cost, measurement range, measurement accuracy, response speed, and other application requirements, a signal following the preamplifier circuit was developed to amplify or reduce the signal, which ensures that the input signal can be adapted to the normal operating range of the microcontroller. The identified signal can be designed into two routes for waveform identification and parameter measurement. One route, a high-performance analog-to-digital converter (ADC), was adopted to quantize the signal to be measured. The waveform and parameters were then identified and measured using a fast Fourier transform (FFT) [15, 16]. The frequency and voltage peak-to-peak values were calculated. Another route was to use a high-performance ADC and direct memory access (DMA) to complete the data transmission. The AD conversion was triggered by a timer, and the waveform duty cycle and other parameters were measured.

The system is operated and controlled by an STM32 microcontroller, and waveform recognition and an online display are used. The current waveform type, duty cycle, peak-to-peak value, frequency, and other important parameters are output in real time. Thus, circuit designers can determine the current state of the signal accurately. In the circuit test, these circuit design methods and spectrum analysis algorithms could identify the waveform and measure the important parameters quickly and accurately, with an accuracy in ± 1% and a response speed of 0.3 s, which meet the requirements of most high-precision and fast waveform measurement applications.

## System and methods

Accurate waveform-type identification and high-precision parameter measurements are important for signal processing. In particularly, such applications as medical devices in the life sciences and radar satellites in the military require support from information-processing technologies [17]. Test and measurement instruments are also important judgment bases for information technology to meet precision requirements. At present, international test and measurement instruments mainly come from several international instrument companies. There are many functions, including high resolution and wide bandwidth. Other spectrum-analyzing equipment on the market is generally expensive. However, for commonly used signal recognition and analysis instruments, on the one hand, there is no requirement for high precision; on the other hand, cost is an important consideration. With the continuous development of information technology, signal analysis hardware and software are relatively mature, and the use of existing equipment can be considered to build high-precision, easy-to-operate, cost-controllable signal measurement equipment through synergistic hardware and software calculations and debugging to complete the rapid identification of the signal and the accurate measurement of important parameters [18].

Theoretically, any continuously measured time-domain signal can be represented as an infinite superposition of sine wave signals of different frequencies. Using the Fourier transform, the RMS values of the fundamental and multiple harmonic components of a non-sinusoidal waveform are calculated, assuming that the signal to be sampled is a periodic function of time, whose expression is

$$x(t) = \sum_{n=0}^{\infty}[bn\cos n\omega t + an\sin n\omega t], \ u_i = U_i\sin\omega t \tag{1}$$

$$a_n = \frac{2}{T}\int_0^T u(t)\sin n\omega_1 t dt, \ b_n = \frac{2}{T}\int_0^T u(t)\cos n\omega_1 t dt \tag{2}$$

Its discrete expression is

$$x[n] = \frac{1}{N}\sum_{k=0}^{N-1} X[k]e^{j(\frac{2\pi}{N})kn} \tag{3}$$

where N is the number of sampling points per cycle, $X_k$ is the kth sampling result, and $X_0$ and XN are the 0th and Nth sampling results, respectively. Therefore, the base wave RMS is

$$U_1 = \sqrt{\frac{a_1^2 + b_1^2}{2}} \tag{4}$$

Here, the initial phase angle is

$$\phi = tg^{-1}(b_1/a_1) \tag{5}$$

The total harmonic RMS value is the RMS of the RMS values of the individual harmonic components:

$$U = \sqrt{u_1^2 + u_2^2 + u_3^2 + \ldots + u_N^2} \tag{6}$$

Waveform recognition is used to determine the signal waveform in the frequency domain. A sine wave has only a fundamental component, whereas a rectangular and a triangular wave have harmonic components, such as the third, fifth, and seventh harmonics, in addition to the

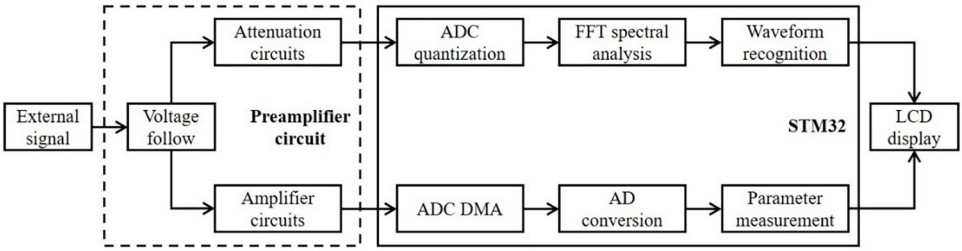

**Fig 1. Block diagram of the system structure.**

fundamental components, and the third-harmonic components are 1/3 and 1/9, respectively. The third-harmonic components are 1/3 and 1/9 of the fundamental, and the ratio of the harmonic RMS values can also be used as a reference for waveform recognition.

The waveform recognition and measurement system in this study can be divided into two parts. The preamplifier circuit is responsible for voltage following, including amplification and attenuation circuits to ensure that the microcontroller can work properly within the acceptable voltage range. It is also necessary to ensure that seamless switching is maintained in low-to-medium-voltage and high-to-medium-voltage situations. The second part is the microcontroller signal-processing module. The two signals sent from the preamplifier circuit are analyzed spectrally one way to identify the waveform and measure the peak-to-peak value of the voltage. The other way is transmitted by the ADC DMA to measure the duty cycle and frequency of the waveform through the timing circuit. A block diagram of the system is shown in Fig 1.

For periodic waveforms, including positive and negative half-weekly signal cycle changes, the direct reading signal of the microcontroller should be greater than zero. Therefore, it is necessary to use a bias circuit to convert the negative half-week signal into a positive half-week waveform to ensure that the microcontroller can read the signal accurately. The parallelism of the calculation is improved to ensure the speed of the parameter measurement. The signal is first read dynamically, and then a voltage-following strategy is adopted to attenuate the signal for high-voltage-value input (>5 V) and amplify the signal for low-voltage-value input (<2 V). Finally, the signal is quantized by the ADC, one way using Fourier transform for spectrum analysis to identify the waveform and measure the frequency, and the other way through the difference between the over-zero voltage value and the maximum voltage value to measure the peak-to-peak value and calculate the duty cycle.

To measure a typical large-span voltage range, a value of 50 mV to 10 V must be measured. Because the input voltage range of common ADCs is 0–2.5 V, the signal to be measured must be attenuated or amplified to ensure the proper operation of the ADC. Usually, the signal modulation circuit and the linearity of the ADC work better in the middle region than at the ends. Here, 2 V is used as the upper measurement limit, and the lower measurement limit is set to 200 mV. This method ensures that the signal in the low-voltage region can be amplified and seamlessly connected to the middle-voltage region. Moreover, for the high-voltage case of 5–10 V, signal attenuation is required to ensure a seamless connection with the medium-voltage region. The medium voltage is set to 2 V. The schematic diagram of the preamplifier circuit section is shown in Fig 2.

The essential technologies for high-precision waveform synchronization recognition and measurement systems include waveform recognition and parameter measurement. Based on the preamplifier circuit, part of the signal sampling is performed by the high-performance ADC of the microcontroller to quantize the signal to be measured. A sufficiently high sampling rate is an important factor in ensuring the accuracy of waveform recognition and

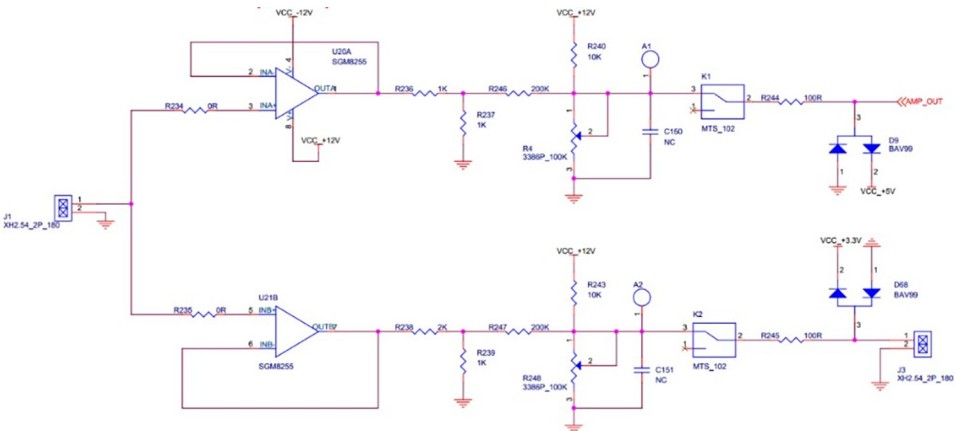

**Fig 2. Schematic diagram of the front circuit section.**

measurement. On this basis, a Fourier transform is used for spectrum analysis to identify the waveform type. The other part transfers data through the ADC DMA, starts the TIM3 clock departure AD conversion, and collects important parameters, such as the peak-to-peak value, frequency, and duty cycle of the current waveform.

The program design process has two steps. One is waveform identification and frequency, and peak-to-peak measurements. First, high-frequency ADC sampling is used to determine whether the signal is complete, sample the complete signal, access the signal rising edge/falling edge time, calculate the frequency of the waveform changes, and stop the capture channel signal sampling clock. After the frequency has been calculated, the complete signal flag is reset, and the timer is turned on to check whether the DMA reception is complete; in this case, the DMA directly accesses the data and filters out the maximum and minimum values. The peak-to-peak value is calculated from the difference. Subsequently the Fourier transform is used to obtain the values of the first- and second-harmonic components, calculate the ratio of the second-harmonic component to the first-harmonic component, calculate the slope, and identify the waveform. Finally, the DMA flag is reset to ensure data capture for the other timer channel. This is shown in Fig 3(A).

In the next step, to measure the duty cycle, the time between two timing edges is calculated. When timer TIM5 starts to capture data, the system determines whether it is the first rising edge of the signal, is the first time of data initialization, and then determines the falling edge. The high-voltage time is calculated while the signal integrity is detected. The low-voltage time and the ratio between the two are calculated to derive the signal duty cycle, and finally the data are cleared. Here, a microcontroller with fast calculation ability is needed, such as the STM32F4, which can reach approximately 170 MHz to meet the current demand for fast calculation, as shown in Fig 3(B).

As shown in Fig 3(A), after starting high-speed ADC sampling, it is first necessary to detect whether the signal is complete and in a complete cycle signal before the frequency can be calculated accurately. After the DMA captures the data, the maximum and minimum values are screened, and the peak-to-peak value is calculated from their differences. In addition, the FFT is used to derive the value of each harmonic component. Through the ratio of the first- and second-harmonic components, the type of waveform is calculated, judged, and simultaneously given to the display part of the microcontroller. The spectrum is analyzed. The results are shown in Fig 4.

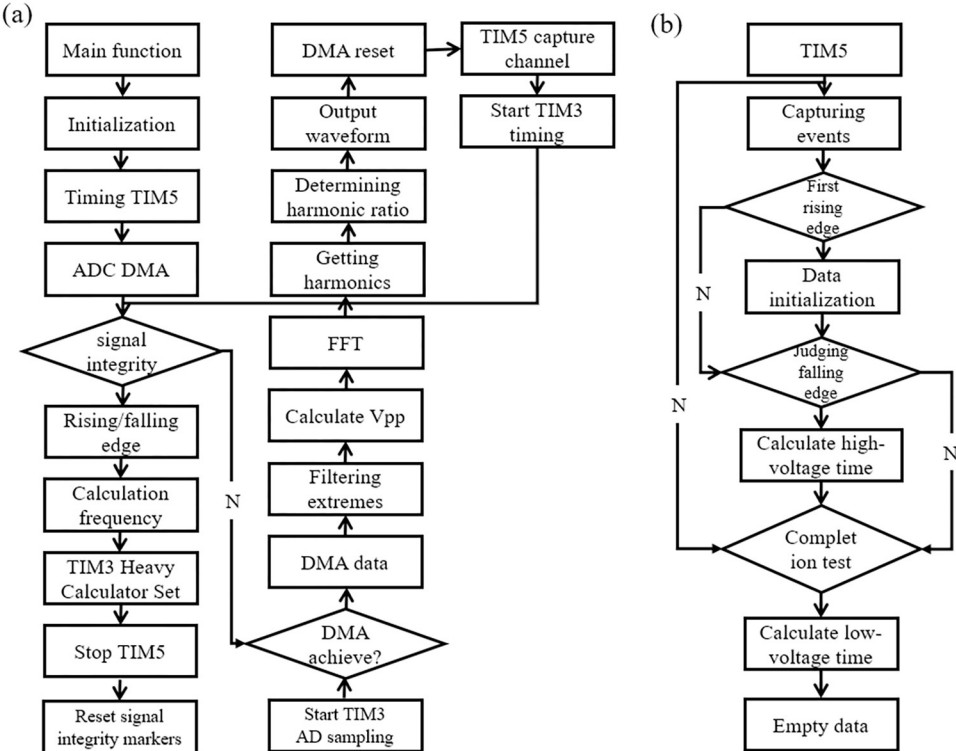

**Fig 3. Flowchart of program execution.** (a) waveform identification and frequency and peak-to-peak value measurement and (b) duty cycle measurement.

## Results and discussion

To test the measurement accuracy, range, and response speed of this high-precision waveform synchronization recognition and measurement system, a random waveform is fed by a signal generator, and the system recognizes the waveform and calculates the frequency, peak-to-peak value, and duty cycle, with the expected response time timed within 0.3 s. Two sets of waveforms with peak-to-peak values and frequencies of $1\ V \leq V_{PP} \leq 5\ V$, $100\ Hz \leq f \leq 10\ kHz$, and $50\ mV \leq V_{PP} \leq 10\ V$, $1\ Hz \leq f \leq 50\ kHz$ are given by the signal generator and passed through the waveform measurement system for identification and parameter measurement. The test results in the range of $1\ V \leq V_{PP} \leq 5\ V$, $100\ Hz \leq f \leq 10\ kHz$ are shown in Fig 5(A). Test results of waveforms in the range of $50\ mV \leq V_{PP} \leq 10\ V$, $1\ Hz \leq f \leq 50\ kHz$ are shown in Fig 5(B). The waveform identification and parameter measurement response times for both are approximately 0.2 s. The response times of the waveform identification and parameter measurement for both are approximately 0.2 s.

Based on the waveform test results shown in Fig 5(A), the original signal has a frequency of 1 kHz and a peak-to-peak value of 2 V with a 50% duty cycle sine wave. After passing through the high-precision waveform synchronization recognition and measurement device, the displayed results show a sine wave with a frequency of 1001 Hz, a peak-to-peak value of 1941.1 mV, and a duty cycle of 50.5%. The accuracy of the waveform recognition is evident, with a frequency error of only 0.1%, peak-to-peak measurement error of 2.9%, and duty cycle error of 1%. Fig 5(B) shows that the original signal is a sine wave with a frequency of 1 kHz and a peak-to-peak value of 10 V. After passing through a high-precision waveform synchronization recognition and measurement device, the displayed result is a sine wave with a frequency of

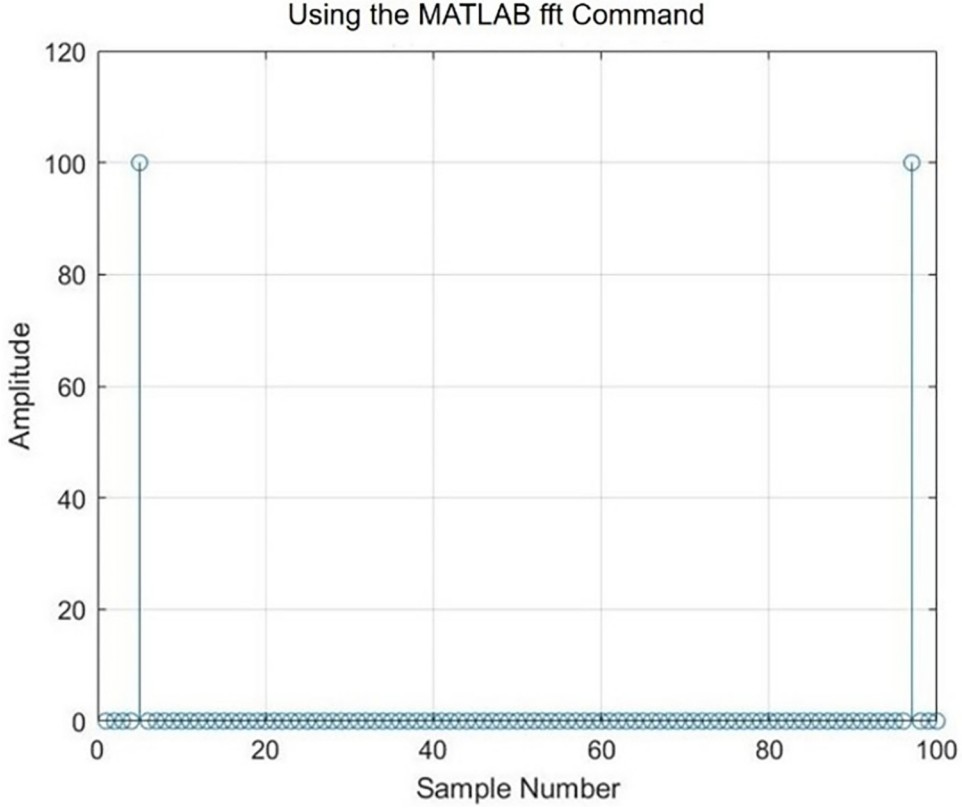

**Fig 4. Spectrum analysis of FFT.**

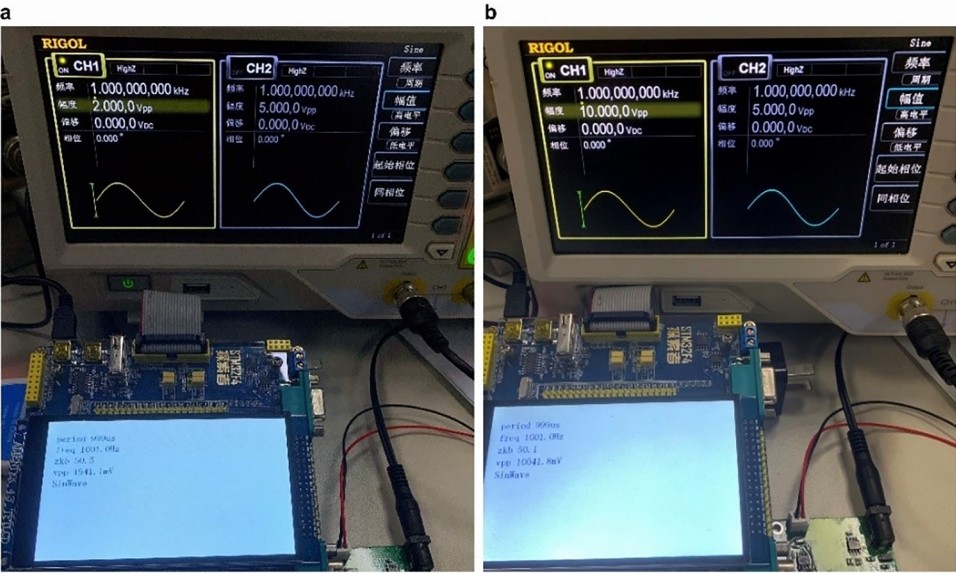

**Fig 5. Waveform test results.** (a) test results in the range of 1 V $\leq$ V$_{PP}$ $\leq$ 5 V, 100 Hz $\leq$ f $\leq$ 10 kHz and (b) test results in the range of 50 mV $\leq$ V$_{PP}$ $\leq$ 10 V, 1 Hz $\leq$ f $\leq$ 50 kHz.

1001.0 Hz, peak-to-peak value of 10041.8 mV, and duty cycle of 50.1%. Waveform recognition is accurate, with a frequency error of only 0.1%, peak-to-peak measurement error of 2.1%, and duty cycle error of 0.2%. The measurements listed above fall within the acceptable ±3% error range for high-precision electronic products. Nevertheless, for irregular or nonperiodic input signals, the current methods may reduce the measurement accuracy and response speed because of the limited signal judgment method and insufficiently high chip sampling frequency. The replacement of higher-performance chips with smarter recognition methods can be improved.

## Conclusions

The high-precision waveform synchronization recognition and measurement system is a waveform recognition and parameter measurement device for periodic signals designed and fabricated using the STM32F407ZGT6 microcontroller as the core, combined with signal scaling circuits, high-frequency ADC sampling, FFT algorithms, DMA transfers, and other signal-processing technologies. The basic waveform generated by the function signal generator is input directly into the voltage-scaling preamplifier circuit, and signals exceeding the voltage operating range of the microcontroller are attenuated and then entered to the microcontroller. The microcontroller acquires and calculates the data of the input waveform and recognizes the waveform and the measurement and display of the relevant parameters. The circuit simulation and actual verification demonstrate that the absolute value of the relative error between the waveform recognition results and all measured parameters in the peak-to-peak value $V_{PP}$ and frequency f is not greater than 3%, and the absolute value of the absolute error is not greater than 2% when the duty cycle D of the rectangular wave signal is in the range of 20%-80%. The response time of the waveform recognition and parameter measurement system is less than 0.3 s. The intelligent linkage and rapid response are suitable for such applications as fast and high-precision waveform recognition and critical parameters measurements. However, for irregular or non-periodic input signals, the measurement accuracy and response speed will be degraded due to the limited chip performance and judgment method, and the next step will be to replace the high-performance chip and intelligent measurement method to improve the system performance.

## Supporting information

**S1 Data.**
(RAR)

## Author Contributions

**Data curation:** Dan Gui.

**Funding acquisition:** Dan Gui.

**Methodology:** Dan Gui.

**Software:** Zong-jie Yu.

**Writing – original draft:** Dan Gui.

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
