## [Decision Letter · Decision Letter 0]

27 Nov 2023

PONE-D-23-24833Research on the key technology of high precision waveform synchronization identification and measurementPLOS ONE

Dear Dr. gui,

Thank you for submitting your manuscript to PLOS ONE. After careful consideration, we feel that it has merit but does not fully meet PLOS ONE’s publication criteria as it currently stands. Therefore, we invite you to submit a revised version of the manuscript that addresses the points raised during the review process.

**ACADEMIC EDITOR: **The reviewers recommend reconsideration the manuscript with revision and modification. I invite the authors to resubmit the manuscript after addressing the comments raised by the reviewers.

We look forward to receiving your revised manuscript.

Kind regards,

Dhanamjayulu C, Ph.D & Post.Doc

Academic Editor

PLOS ONE

“This work was supported by the industry-University-Research Project of Wuhan Municipal Education Bureau (CXY202014); Education Research Program Project of Hubei Provincial Department (B2020405).”

“the industry-University-Research Project of Wuhan Municipal Education Bureau (CXY202014); Education Research Program Project of Hubei Provincial Department (B2020405)”

Additional Editor Comments:

The reviewers recommend reconsideration the manuscript with revision and modification. I invite the authors to resubmit the manuscript after addressing the comments raised by the reviewers.

Reviewers' comments:

Reviewer's Responses to Questions

**Comments to the Author**

1. Is the manuscript technically sound, and do the data support the conclusions?

Reviewer #1: Yes

Reviewer #2: Yes

2. Has the statistical analysis been performed appropriately and rigorously? 

Reviewer #1: Yes

Reviewer #2: Yes

3. Have the authors made all data underlying the findings in their manuscript fully available?

Reviewer #1: No

Reviewer #2: Yes

4. Is the manuscript presented in an intelligible fashion and written in standard English?

Reviewer #1: Yes

Reviewer #2: No

5. Review Comments to the Author

Reviewer #1: 我仔细阅读了题为“高精度波形同步识别与测量关键技术研究”的手稿，以便可能发表在PLoS One上。

作者开发了一种放大和衰减预选器电路，用于测量各种值的波形。之后，作者采用高精度模数转换和快速傅里叶算法来捕获和分析波形的关键参数。硬件和软件的结合是一个好主意。仿真和实验数据验证了所提电路和快速识别算法的可行性和有效性。这项工作将引起波形识别界的兴趣，特别是那些精度高、响应速度快的应用。在这份手稿在PLoS One中被公众接受之前，必须进行小的修改。

1. 在引言部分，波形识别和参数测量对于应用广泛的信号处理至关重要，作者需要补充相关文献。

2. 图1.系统结构的框图。调整模块大小以获得更好的图形，例如将字母“外部”放在同一行上。

3.限制讨论。所提方法对基本波形判断和计算速度有显著影响。但是，是否也可以准确快速地测量不规则或非周期性的输入信号，可以进一步澄清。

4、注意表达;目前的稿件需要写得简洁明了。

Reviewer #2: Waveform synchronization and measruement is an important branch of instrumentation, while the current manuscript has not presented sufficient proof to reach the publication threshold of Plos One.

Following is some concerns:

1.Frquency error of 0.1% is rather far from an advanced progress, although it depends on measurement time, SNR, etc. To what extend one can push the precision and accuracy, formulus 2 in a RSI paper(doi: 10.1063/5.0002377) has already clarified.

2.Comparing to a lock-in amplifier, mV-range measruement is far from scientific researches, so authors are strongly recommended to upgrade the measurement construction for high precision measurement with lower cost.

3.STM32F4 is relatively slow for electronic feedback, so it is only suitable for some situations where delay of seconds is acceptable. Nevertheless, authors have not clearly shown a neccessary application.

6. PLOS authors have the option to publish the peer review history of their article (what does this mean?). If published, this will include your full peer review and any attached files.

Reviewer #1: No

Reviewer #2: **Yes: **Sigma-Jun Lu

---

## [Author Response · Author response to Decision Letter 0]

2 Feb 2024

Response to reviewers’ comments

Reviewer #1

I have carefully read the manuscript entitled “Research on the key technology of high precision waveform synchronization identification and measurement” for possible publication in PLoS One. 

The authors developed an amplifying and attenuating preselector circuit to measure waveforms across various values. Afterward, the authors adopted high-precision analog-to-digital conversion and a fast Fourier algorithm to capture and analyze the critical parameters of the waveforms. The combination of hardware and software was an excellent idea. Simulation and experimental data demonstrate the feasibility and efficiency of the proposed circuit and fast recognition algorithm. This work will be of interest to the waveform recognition community, especially those applications with high precision and quick response requirements. A minor revision has to be done before this manuscript can be accepted for the public in PLoS One.

Reply: Many thanks for the kind words.

Major comment-1: In the introduction section, waveform identification and parameter measurement are crucial for signal processing with a wide range of applications, and the authors need to add relevant literature. 

Reply: The relevant literature on the wide range of applications of waveform identification and parameter measurement has been added to the revised manuscript to illustrate its important role in the field of signal processing.

Major comment-2: Fig 1. The block diagram of the system structure. Resize modules for better graphics, such as putting the letters "External" on the same line. 

Reply: Thank you so much for your careful check, the overall structure of Figure 1 has been optimized in the revised manuscript. 

Major comment-3: Limitation discussion. The proposed method has significant effects on essential waveform judgment and calculation speed. However, whether the input signals with irregular or non-periodic can also be measured accurately and quickly could be given additional clarification. 

Reply: Thank you very much for your constructive comments. The limitation discussion has been added to the Conclusion and Discussion section of the revised manuscript. For irregular or non-periodic input signals, the method proposed in the manuscript may be reduced in measurement accuracy and response speed due to the limited signal judgment method and insufficiently high chip sampling frequency. The replacement of higher performance chip and smarter recognition method case can be effectively improved，which is the next part of our upcoming research.

Major comment-4: Pay attention to expression; the current manuscript needs to be written concisely and straightforwardly. 

Reply: Language and content have been refined and revised throughout the text in the revised manuscript. 

Reviewer #2

Waveform synchronization and measurement is an important branch of instrumentation, while the current manuscript has not presented sufficient proof to reach the publication threshold of Plos One. Following is some concerns:

Major comment-1：Frequency error of 0.1% is rather far from an advanced progress, although it depends on measurement time, SNR, etc. To what extend one can push the precision and accuracy, formulus 2 in a RSI paper (doi: 10.1063/5.0002377) has already clarified.

Reply: Thank you for your comments. For localized fine frequency measurement, the lock-in frequency measurement method in the paper (doi: 10.1063/5.0002377) has high accuracy and efficiency. But for wide frequency ranges, the FFT-like methods are general, and considering the cost, application range and other factors, the FFT-based measurement methods in this manuscript can meet the requirements of waveform measurement in both the fundamental frequency range and wider frequency ranges.

Major comment-2：Comparing to a lock-in amplifier, mV-range measurement is far from scientific researches, so authors are strongly recommended to upgrade the measurement construction for high precision measurement with lower cost.

Reply: Thanks to these suggestions, this manuscript is aimed at generalized scenarios to achieve arbitrary waveform identification and key parameter measurement. in the next step, we will upgrade the measurement structure with specific applications on low cost and high accuracy.

Major comment-3：STM32F4 is relatively slow for electronic feedback, so it is only suitable for some situations where delay of seconds is acceptable. Nevertheless, authors have not clearly shown a necessary application.

Reply: Considering the cost, universalization and other factors, STM32F4 is a widely used microcontroller that can meet the response speed requirements for waveform recognition and parameter measurement in basic and wider frequency ranges. Though upgrading the microprocessor chip, faster response speed can be realized.

---

## [Editor Report · Decision Letter 1]

5 Feb 2024

Research on the key technology of high precision waveform synchronization identification and measurement

PONE-D-23-24833R1

Dear Dr. gui,

We’re pleased to inform you that your manuscript has been judged scientifically suitable for publication and will be formally accepted for publication once it meets all outstanding technical requirements.

Kind regards,

Dhanamjayulu C, Ph.D & Post.Doc

Academic Editor

PLOS ONE

Additional Editor Comments (optional):

The authors have revised the article as per reviewers concerns.

It can be accepted in present form .
---

## [Editor Report · Acceptance letter]

23 Feb 2024

PONE-D-23-24833R1 

PLOS ONE

Dear Dr. Gui, 

I'm pleased to inform you that your manuscript has been deemed suitable for publication in PLOS ONE. Congratulations! Your manuscript is now being handed over to our production team.

Kind regards, 

on behalf of

Dr. Dhanamjayulu C 

Academic Editor

PLOS ONE